

# The interplay of motor adaptation and groupitizing in numerosity perception: Insights from visual motion adaptation and proprioceptive motor adaptation

Huanyu Yang, Liangzhi Jia, Jun Zhu, Jian Zhang, Mengmeng Li, Chenli Li and Yun Pan

Key Laboratory of Basic Psychological and Cognitive Neuroscience, School of Psychology, Guizhou Normal University, Guiyang, China

## ABSTRACT

Groupitizing is a well-established strategy in numerosity perception that enhances speed and sensory precision. Building on the ATOM theory, Anobile proposed the sensorimotor numerosity system, which posits a strong link between number and action. Previous studies using motor adaptation technology have shown that high-frequency motor adaptation leads to underestimation of numerosity perception, while low-frequency adaptation leads to overestimation. However, the impact of motor adaptation on groupitizing, and whether visual motion adaptation produces similar effects, remain unclear. In this study, we investigate the persistence of the advantage of groupitizing after motor adaptation and explore the effects of visual motion adaptation. Surprisingly, our findings reveal that proprioceptive motor adaptation weakens the advantage of groupitizing, indicating a robust effect of motor adaptation even when groupitizing is employed. Moreover, we observe a bidirectional relationship, as groupitizing also weakens the adaptation effect. These results highlight the complex interplay between motor adaptation and groupitizing in numerosity perception. Furthermore, our study provides evidence that visual motion adaptation also has an adaptation effect, but does not fully replicate the effects of proprioceptive motor adaptation on groupitizing. In conclusion, our research underscores the importance of groupitizing as a valuable strategy in numerosity perception, and sheds light on the influence of motion adaptation on this strategy.

## INTRODUCTION

Numerosity perception is a fundamental perceptual property, akin to color, brightness, size, and motion (*He et al., 2015*; *Grasso et al., 2022*; *Leslie, Gelman & Gallistel, 2008*). This cognitive ability enables the extraction of numerical information from stimuli by human adults, infants, and even animals, facilitating quick estimation, manipulation, and processing of numerosities, and comprehension of their relationships (*Deheane, 1998*; *Clarke & Beck, 2021*; *Crollen & Collignon, 2020*; *Grasso et al., 2022*). Recent studies

Corresponding author
Yun Pan, panyun129@163.com

exploring animal numerosity perception delve into the quantitative assessments made by animals during environmental negotiations. These investigations reveal the engagement of both cortical and subcortical brain regions in magnitude estimation, suggesting a more extensive neural network at play (*Vallortigara et al., 2022*). Another line of research provides a comprehensive review of numerosity perception across various animal species, underscoring the necessity of comprehending the neural mechanisms underpinning observed behavioral patterns (*Lorenzi, Perrino & Vallortigara, 2021*). Furthermore, a study focusing on vertebrates, particularly fish, showcases their adeptness at estimating quantities, shedding light on the neurobiological foundations that support this cognitive capacity (*Messina et al., 2022*). These studies indicate that both humans and other species are capable of perceiving and processing numerical information in their environment. Some researchers have suggested that individuals, both human and animal, draw inferences based on life experiences in numerosity judgment, incorporating non-numerical characteristics of stimuli such as color, size, and orientation (*Grasso et al., 2022*; *Dehaene, Bossini & Giraux, 1993*; *Zorzi, Stoianov & Umiltà, 2005*). Hence, the investigation of numerosity perception necessitates a multidimensional approach, linking numerical quantity or numerosity with diverse perceptual features.

One of the prominent theories of the interaction of numerosity and other perceptual features is "A Theory of Magnitude" (ATOM). ATOM suggests that, space, time, and number are processed through a common magnitude system and are linked by the action. The parietal cortex is an important brain area for magnitude system processing, to optimize action programming and execution (*Walsh, 2003*). ATOM exhibits species generality, as it observed not only in humans but also prevalent across various animal species. Initially discovered in mammals, it was linked to the mediation by the parietal cortex. However, recent research indicates that this theory extends beyond mammals to encompass a broader range of animals. For instance, studies in birds, despite lacking a cortex akin to mammals, reveal reliable space–time interactions, supporting the cross-species applicability of ATOM (*De Corte, Navarro & Wasserman, 2017*). Insects, such as bees, demonstrate numerical discrimination, and training experiments illustrate a cross-dimensional transfer between numerical and spatial dimensions. This provides compelling evidence for the universal coding of general magnitude (*Bortot, Stancher & Vallortigara, 2020*). Therefore, the omnipresence of magnitude theory in animals establishes a unified cognitive framework for comprehending their perception of time, space, and quantity. The widespread applicability of the theory underscores a common neural basis that transcends species differences. Research on ATOM mostly focused on the shared mechanism of number and other perceptual features, but little attention has been paid to how quantity information combines with action planning and execution. *Anobile et al. (2021a)* and *Anobile et al. (2021b)* proposed a "sensorimotor numerosity system" that can simultaneously process perception and action planning or execution. The sensorimotor numerosity system possesses the ability to estimate the numerical value of externally induced events as well as the magnitude of internally initiated actions. Additionally, this system is capable of processing quantitative information pertaining to non-numerical attributes such as spatial and temporal dimensions (*Anobile et al., 2021a*; *Anobile et al.,*

*2021b*). In recent years, more and more research has proposed numerosity perception is intrinsically linked with action, as evidenced by adaptation techniques (*Burr et al., 2021*).

Perceptual adaptation, known as "psychologist's micro-electrode", is one of the most effective methods to investigate perceptual mechanisms. The efficacy of this tool as a fundamental psychophysical instrument for investigating numerous perceptual properties has been empirically established (*Thompson & Burr, 2009*), numerosity, as a primary perceptual property, can also be adapted (*Burr & Ross, 2008*). Recent empirical investigations employing motor adaptation techniques have successfully unveiled a robust association between motion and the numerosity estimation (*Burr et al., 2021*; *Anobile et al., 2021a*; *Anobile et al., 2021b*; *Anobile et al., 2016*; *Togoli et al., 2020*). Participants were asked to tap in midair with their dominant hand, either very fast or slow. It was found that fast tapping caused participants underestimate of the numerosity perception, while slow tapping caused overestimation. The motor adaptation has a strong effect, causing a deviation of about 20% to 25% in the numerosity perception. This effect is cross-modal and exists in both the temporal sequences and spatial arrays (*Arrighi, Togoli & Burr, 2014*; *Anobile et al., 2016*; *Maldonado Moscoso et al., 2020a*; *Maldonado Moscoso et al., 2020b*). The motor adaptation effect on numerosity perception is not limited to vision; it is also present in sequences of auditory tones. Even more intriguingly, motor adaptation similarly affects auditory numerosity in congenitally blind adults, indicating that the interaction between action and numerosity perception remains intact irrespective of visual input or even prior visual experience (*Togoli et al., 2020*). Additionally, there is also brain imaging evidence that reveals a link between numerosity perception and motion. The activated regions involved in the process of numerosity perception are located near the parietal cortex and partially overlap with the action-related brain regions (*Hubbard et al., 2005*; *Simon et al., 2002*).

People will use multiple strategies according to their needs when processing numerosity, such as fast and accurate processing of small numerosities within the subitizing range (*Mandler & Shebo, 1982*); fast but rough approximate estimation (*Zhou et al., 2016*); accurate but time-consuming counting (*Simon & Vaishnavi, 1996*); and the groupitizing strategy, which has attracted more and more attention in recent studies. "Groupitizing" means that when estimating the number of an array, dividing the array into several subgroups, the estimated result is faster and more accurate than the ungrouped one (*Pan et al., 2021*; *Anobile et al., 2020a*; *Anobile et al., 2020b*; *Maldonado Moscoso et al., 2020a*; *Maldonado Moscoso et al., 2020*; *Starkey & McCandliss, 2014*). Groupitizing combines the advantages of subitizing and estimation. When the grouped subgroups and the number of items in each subgroup are within the subitizing range, the subgroups and the number of items in each subgroup can be processed simultaneously. Groupitizing is like processing several arrays within the subitizing range at the same time. Therefore, the processing is faster and more accurate than that of non-grouped arrays (*Wege, Trezise & Inglis, 2021*). In addition, using the grouping strategy involves advanced cognitive processing, such as mental arithmetic (*Maldonado et al., 2021*; *Anobile et al., 2020a*; *Anobile et al., 2020b*; *Ciccione & Dehaene, 2020*; *Maldonado Moscoso et al., 2020a*; *Maldonado Moscoso et al., 2020*), making the result of numerosity perception more accurate. The grouping effect is also cross-modal
and cross-format. Groupitizing is found in both time and space dimensions (*Anobile et al., 2021a*; *Anobile et al., 2021b*), and not only visually grouped array have the grouping advantage, but also the auditory tones sequence (*Anobile et al., 2021a*; *Anobile et al., 2021b*).

Perceptual adaptation typically elicits erroneous perception of specific aspects of stimuli, such as motion aftereffect, exhibiting a bias in the direction opposite to that of the adaptation. One of the most prototypical examples is the waterfall illusion: after observing the downward motion of a waterfall for several seconds, redirecting one's gaze to the adjacent region induces an illusory perception of upward motion (*Thompson & Burr, 2009*; *Anobile et al., 2021a*; *Anobile et al., 2021b*; *Barlow & Hill, 1964*; *Addams, 1833*). This phenomenon can be ascribed to the disparity in relative neuronal activities resulting from the continuous stimulation of neurons specialized for vertical motion, leading to adaptation and subsequent imbalance with neurons specialized for downward motion. As a result, when a test stimulus is presented, it makes use of an unbalanced system, which produces measurable illusion perception aftereffects (*Solomon & Kohn, 2014*). Adaptation is a pervasive phenomenon observed across various sensory systems and serves as a transient form of plasticity that can be leveraged as a valuable mechanism for recalibrating the perception of environmental statistics (*Webster, 2015*; *Anobile et al., 2021a*; *Anobile et al., 2021b*). Therefore, visual motion adaptation also induces perceptual adaptation. Previous research has demonstrated that adapting to fast or slow proprioceptive hand movements leads to subsequent deviations in numerosity estimation, thus resulting in an adaptation effect. Regarding visual motion adaptation, only a few studies have explored it (*Fornaciai, Togoli & Arrighi, 2018*). Previous research has laid the groundwork for our understanding of visual motion adaptation. Building upon this foundation, our study aims to further investigate whether motion adaptation effects occur when observing objects with different speeds. Additionally, we will introduce grouping strategies to explore the interplay between these two factors. This endeavor contributes to constructing a more comprehensive theoretical model in the field of numerosity perception. Consequently, this study expands upon the proprioceptive motion adaptation paradigm utilized by prior researchers and incorporates a visual motion adaptation paradigm. Its objective is to examine whether participants, after being exposed to arrays of rapidly or slowly moving stimuli, demonstrate a adaptation effect in numerosity estimation.

The investigation of the interplay between numerosity perception and motor adaptation constitutes a significant domain deserving of scholarly attention, however, the cognitive mechanism of numerosity perception under visual motion adaptation has not been studied in-depth. "Groupitizing" can improve the sensory precision of numerosity perception, while motor adaptation can bias the results: fast or high-frequency motor adaptation leads to underestimation, while slow or low-frequency motor adaptation leads to overestimation. So, an intriguing question is, after the motor adaptation, the next numerosities is presented in groups, then what will be the result of numerosity perception? Therefore, the purpose of this study is to explore whether motor adaptation affects the groupitizing under two motor adaptation conditions or whether there is still an adaptation effect if the groupitizing strategy is used. Based on this, we designed two experiments: Experiment 1 adopts proprioceptive motor adaptation (hand tapping) similar to previous studies, and Experiment 2 adopts

adaptation to visual motion (viewing moving dots) to explore the relationship between motor adaptation and grouping effect. Since many previous studies have proven that the groupitizing strategy has a robust advantage in the numerosity perception, here, we assume that even with motor adaptation before numerosity estimation, the grouping effect still exists. In other words, grouping can reduce the perception deviation after motor adaptation, making the result of numerosity perception more accurate.

## EXPERIMENT 1

### Participants

A total of 26 undergraduate students participated in the study, of which 16 were female, with a mean age of 21.69 years (SD = 1.76). One subject dropped out of the experiment and was excluded, resulting in a final sample size of 25 participants. All participants included in the study exhibited normal or corrected visual acuity and were right-handed. In adherence to ethical guidelines, informed consent was obtained from all participants involved in this study. Detailed information regarding the study's purpose, procedures, and potential risks and benefits was provided to each participant. The consent process took place through face-to-face meetings, allowing participants the opportunity to ask questions before expressing their consent. Written consent was obtained from each participant, and a copy of the consent form was provided to them for their records.

### Methods and procedures

#### Stimuli

The stimulus material was created in accordance with our previous research (*Pan et al., 2021*). The stimulus arrangement consisted of a grid measuring 6° × 6°, composed of 144 squares, each measuring 0.4° × 0.4°. The stimuli were positioned at the intersections of the grids, resulting in a total of 121 possible positions for the stimulus array (see Fig. 1A). In the grouping condition, the stimuli were divided into four groups, with each group having 12 possible positions. The stimuli within each group were distributed within the same quadrant. The grouping conditions included 2-4 subgroups for each numerosity, with each subgroup containing 2-6 items (Fig. 1A). The numerosity range was 5-17, and there were 13 different numerosities used in the study, with the following combinations: N5 (2, 2, 1); N6 (3, 3); N7 (3, 3, 1); N8 (2, 2, 2, 2); N9 (3, 3, 3); N10 (3, 3, 3, 1); N11 (3, 3, 3, 2); N12 (3, 3, 3, 3); N13 (5, 5, 3); N14 (4, 4, 3, 3); N15 (4, 4, 4, 3); N16 (4, 4, 4, 4); N17 (5, 4, 4, 4). In the no-grouping condition, each stimulus was randomly distributed within the large grid (see Fig. 1B).

#### Procedures

Participants sat 60 cm from a screen monitor (refresh rate: 60 Hz) in a quiet, dimly lit room. Stimulus materials were created and presented using E-Prime 3.0 (https://pstnet.com/products/e-prime/). Each trial commenced with a 6-second adaptation screen. Participants were instructed to tap with their dominant hand (right) on the right side of the screen in mid-air as quickly as possible during the high-frequency adaptation level. During the low-frequency adaptation level, participants were instructed to tap at a much

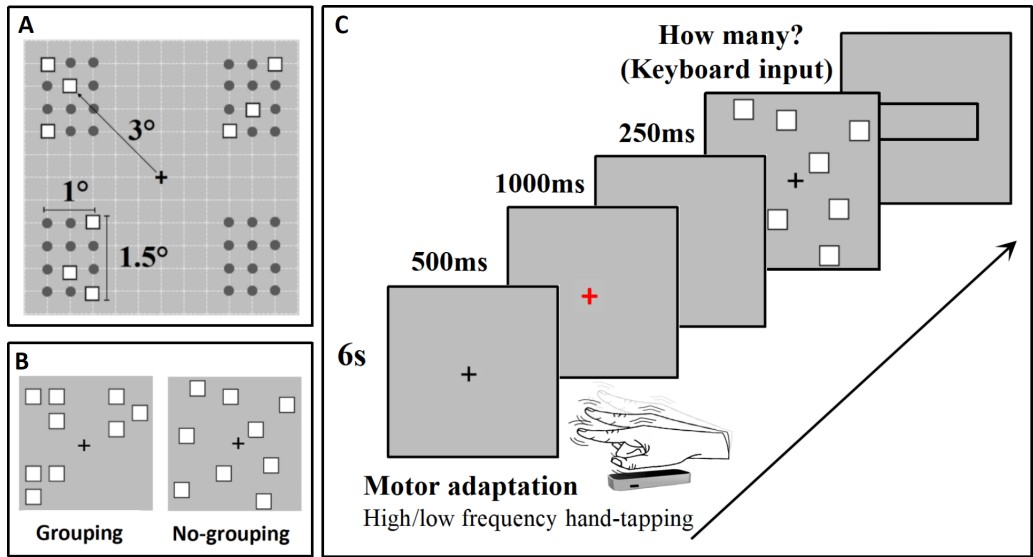

**Figure 1 Stimuli and experimental procedure employed in this study.** (A) The distribution grid used to arrange the stimuli for the grouping condition within the experimental paradigm is displayed; (B) Presents an example of the stimuli utilized in this experiment (Stimuli are not depicted to scale); (C) Illustrates the experimental procedure in detail, providing a comprehensive overview of the steps taken during the study.

slower frequency of approximately 1 Hz (once per second) within the 6-second duration. The two adaptation levels were tested separately. According to the research by *Anobile et al. (2020b)*, the motor adaptation effect occurs within a spatially specific range in external (rather than body-centered) coordinates, confined to a 10° distance centered on the tapping hand. Therefore, in this study, the distance for visual stimuli and adaptation regions is also chosen to be 10°. Hand movement was recorded and monitored by infrared motion sensor equipment (Leap motion controller: https://www.leapmotion.com/). Subsequently, a red "+" appeared in the center of the screen for 500 ms, and participants were instructed to stop tapping when the red "+" appeared. This was followed by a blank screen lasting 1000 ms to allow sufficient time for the participants to stop moving. The stimulus was then presented for 250 ms. After the stimulus disappeared, a rectangular input box appeared, and participants were asked to estimate the number of stimuli and use the numeric keypad to quickly and accurately input the estimated result into the input box (Fig. 1C). The grouping and no-grouping conditions were randomly presented within the same block.

### Statement

All coauthors agreed with the contents of the manuscript. This study was performed in line with the principles of the Declaration of Helsinki. This study was approved by the School of Psychology Ethics Committee at Guizhou Normal University (Approval number: GZNUPSY202111005). The complete raw data, materials, and code used in this experiment are accessible on our Zenodo page: https://doi.org/10.5281/zenodo.10421155. We express our gratitude to an anonymous reviewer for providing this valuable suggestion. All data generated or analyzed during this study have been included in this published article.

## Data analysis

For the data analysis of grouping effects, reaction times (RTs) and coefficients of variation (CVs) were calculated for each participant. CV represents the sensory noise involved in numerosity perception studies and is used to measure variation in the data. An increased CV value indicates greater sensory noise, resulting in less precision in estimations (Eq. (1)). Statistical significance was determined using a $2 \times 2 \times 13$ repeated measures ANOVA with test grouping condition (two levels for grouping: grouping and no-grouping), adaptation level (low frequency and high frequency), and numerosity (13 levels ranging from five to 17) as the main factors. Each numerosity was presented three times under each condition, yielding a total of 156 trials per participant.

$$CV = \frac{i}{N_i} \tag{1}$$

Where $N_i$ is the analyzed numerosities, $N_i = 13$ (ranging from five up to 17), $i$ is the standard deviation of the numerosities.

As for the analysis of adaptation effects, we referred to the 2014 study conducted by Arrighi and colleagues (*Arrighi, Togoli & Burr, 2014*), utilizing the Adaptation Index (AI) to quantify the adaptation magnitude. The adaptation effect is operationally defined as the difference between perceived numerosity following low-frequency adaptation and those following high-frequency adaptation. To quantitatively assess the extent of adaptation, we employed linear regression, fitting the differential curve of adaptation amplitude and calculating its slope. The multiplication of this slope by 100 yields the Adaptation Index (AI), providing an estimate of the magnitude of adaptation (*Arrighi, Togoli & Burr, 2014*). To assess the statistical significance of the variances in adaptation effects between grouping and no-grouping condition, a two-tailed paired $t$-test was utilized (*Anobile et al., 2016*).

## Results

### Effects of adaptation

The adaptation effect is measured by adaptation index. We calculated the adaptation index separately for the grouping and no-grouping conditions. The results are presented in Fig. 2A, which illustrates the adaptation index under both grouping and no-grouping conditions. Notably, the grouping condition exhibits a more pronounced adaptation effect compared to the no-grouping condition ($t = 3.233$, $p = 0.23$, Cohen's $d = 1.572$). Fig. 2B illustrates the mean perceived numerosity (averaged across all subjects) plotted against the physical number of pulses. The mean estimates without adaptation, represented by the gray dotted line, closely aligned with the veridical values. A zero-anchored linear regression model ($R^2 = 0.99$) effectively described the data, exhibiting a best-fitting slope of 0.99. The introduction of high-frequency adaptation consistently led to a 9% reduction in apparent numerosity across all tested numerosities (linear regression slope of 0.90), as can be seen from Fig. 2B, the line shifted towards the upper side compared to the baseline (no adaptation), indicating an overestimation of perceived numerosity. Conversely, low-frequency adaptation resulted in a 4% increase in numerosity (linear regression slope of 1.03, compared to the baseline slope of 0.99), the line shifted towards the upper side compared to the baseline (no adaptation), indicating an overestimation of perceived

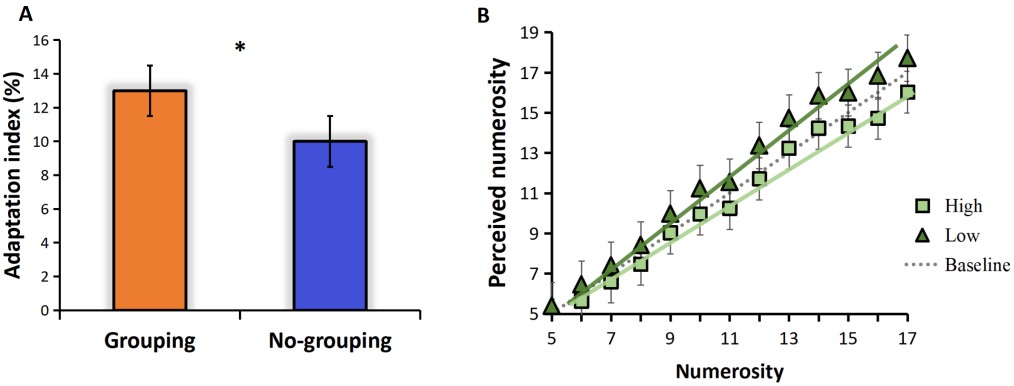

**Figure 2** **Results.** (A) Adaptation index under grouping and no-grouping conditions; (B) The perceived numerosity, averaged across trials and subjects, varies as a function of physical numerosity in the three adaptation conditions. The analysis includes best-fitting linear regressions ($R^2 > 0.98$ in all conditions). The regression slopes are as follows: no adaptation (dotted line) = 0.99; high-frequency adaptation (square) = 0.90; low-frequency adaptation (triangle) = 1.03 ($p < 0.001$ in both conditions).

numerosity (Fig. 2B). Given that the zero-anchored linear regressions explained over 98% of the variance across all conditions, it suggests that adaptation influenced numerosities uniformly across the entire range.

### Groupitizing and adaptation effect

We analyzed the coefficient of variation (CVs) under the conditions of high and low frequency motor adaptation. The statistical analysis employed in this study involved conducting a repeated measures analysis of variance (ANOVA), which revealed a significant main effect of adaptation level, $F(1,24) = 13.128$, $p = 0.001$**, the high frequency hand tapping was more accurate and had less sensory noise than low frequency hand tapping conditions (Fig. 3A). The main effect of numerosity was also significant, and the sensory precision of small numerosities was more accurate. Different from previous studies, no grouping advantage was found in this study, that is, the main effect of group conditions was not significant, $F(1,24) = 0.599$, $p = 0.447$.

For the ANOVA on RTs, We found a significant main effect of adaptation level, $F(1,24) = 83.004$, $p < 0.001$***, the reaction time of the high frequency hand tapping was shorter than that of low frequency hand tapping conditions. A significant main effect of group condition was found, $F(1,24) = 184.833$, $p < 0.001$***, The grouping condition resulted in faster reaction times compared to the no-grouping condition. There was also a significant main effect of numerosity, $F(12,23) = 2.657$, $p = 0.047$*, small numerosities react faster than large numerosities. And the interaction between adaptation level and grouping was significant, $F(1,24) = 13.641$, $p = 0.001$**. Due to the significant interaction, we conducted additional simple effects analyses to examine differences in grouping conditions under different adaptation frequencies. The results revealed that high-frequency adaptation exhibited a stronger grouping effect, with grouped conditions showing faster reaction times compared to no-grouping conditions under high-frequency adaptation (Fig. 3B).

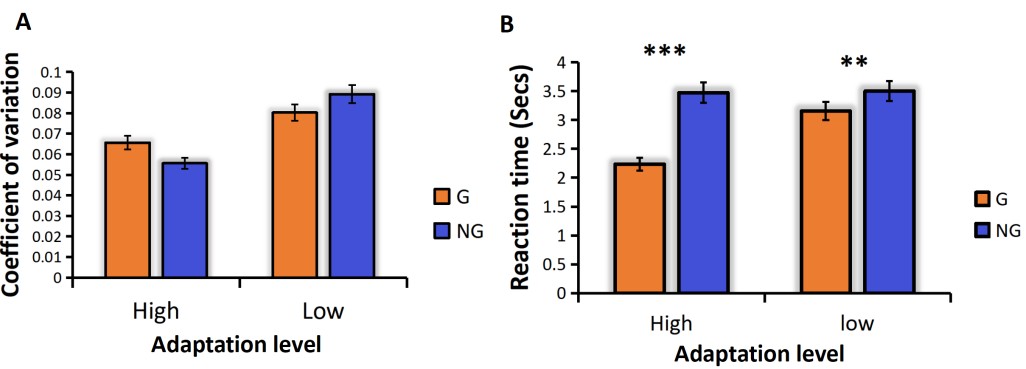

**Figure 3** (A) CVs for different adaption level by group condition. (B) RTs for different adaption level by group condition. "G" for grouping condition, and "NG" for no-grouping condition. ***$p < 0.001$; **$p < 0.01$; *$p < 0.05$.

## Discussion of Experiment 1

In Experiment 1, our findings were consistent with previous studies (*Anobile et al., 2016*; *Anobile et al., 2020a*; *Anobile et al., 2020b*), demonstrating that the adaptation to high frequency hand tapping was underestimated in the subsequent numerosity estimation task, while the adaptation to low frequency hand tapping was overestimated (Fig. 3A). These results indicate that the adaptation effect persists even when subjects adopt the "groupitizing" strategy. Previous research has established that visually grouped arrays tend to result in faster and more accurate numerosity perception compared to ungrouped arrays, a phenomenon referred to as "groupitizing" (*Anobile et al., 2020a*; *Anobile et al., 2020b*; *Ciccione & Dehaene, 2020*). However, contrary to these findings, our results did not reveal a significant grouping effect on perception precision. The sensory precision under the grouping condition was not significantly higher than that under the non-grouping condition (Fig. 3A), the reason for this result is that the grouping effect is affected by the motor adaptation, so that no grouping advantage is found. Interestingly, our data did show a grouping advantage in reaction time. As shown in Fig. 3B, the reaction time in the grouping condition was significantly faster than that in the no-grouping condition.

Experiment 1 further reveals a stronger adaptation effect under grouping conditions. As illustrated in Fig. 2A, the adaptation index in the grouping condition is significantly higher than that in the no-grouping condition, indicating a more pronounced deviation in numerosity estimation induced by motor adaptation under grouping conditions. This finding contradicts our initial expectations, as we anticipated that grouping conditions might enhance the precision of numerosity perception. Instead, it resulted in a greater estimation bias under the influence of motor adaptation.This unexpected observation prompts a deeper consideration of the interactive mechanisms between motor adaptation and grouping effects. One possible explanation is that motor adaptation may alter the weighting allocation for processing grouping information, making the adaptation effect more prominent. On the other hand, motor adaptation and groupitizing may involve distinct neural mechanisms, which could lead to mutual competition or enhancement
effects in numerosity perception. Furthermore, our study results serve as a reminder that numerosity perception is a complex process influenced by multiple factors, to comprehensively understand the neural underpinnings of numerosity perception, future research can delve into exploring the interactions among these factors in greater detail.

# EXPERIMENT 2

In the field of motion perception, prior research has predominantly focused on proprioceptive motor adaptation, such as hand tapping, while adaptation to visual motion, such as viewing a motion lattice, has received limited attention. Despite the growing body of literature on motor perception (*Anobile et al., 2021a*; *Anobile et al., 2021b*; *Sixtus et al., 2023*; *Anobile et al., 2016*), however, as of now, there has been no research exploring the relationship between visual motion adaptation and the grouping effect. Hence, Experiment 2 in our study fills this research gap by investigating the potential interaction between motion adaptation and grouping strategies during visual motion adaptation. This novel contribution to the literature can enhance our understanding of the underlying mechanisms of motion adaptation and its relationship with grouping strategies.

## Participants

We collected data from 24 undergraduate students form Guizhou Normal University (15 Female; Mean age = 22.35; SD = 2.33). None of them participated in the experiment 1. All recruited participants had normal or corrected visual acuity, all right-handed and signed an informed consent form.

## Methods and procedures

### Stimuli

Stimuli were created and presented using E-prime 3.0. In Experiment 1, proprioceptive motor adaptation of hand tapping was used as the motor adaptation stimulus, while in Experiment 2, visual motion adaptation was used, which involved watching a lattice with fast (3 m/s) or slow (0.5 m/s) movement. A total of 10 blue dots (RGB: 0; 0; 255) were presented, each measuring 0.1° × 0.1°, with random motion direction. Other materials used for stimulation were the same as in Experiment 1.

### Procedures

At the beginning of each trial, a 6-second motor adaptation was performed by watching the lattice with irregular motion at either fast or slow speed, followed by a fixation point "+" displayed for 500 ms. After the disappearance of the fixation point, the stimulation was presented for 250 ms. Participants were then required to estimate the number of stimulations and input the result into the presented input box using a keyboard, followed by pressing "Enter" to proceed to the next trial (Fig. 4).

## Data analysis

Data analysis was the same as experiment 1.

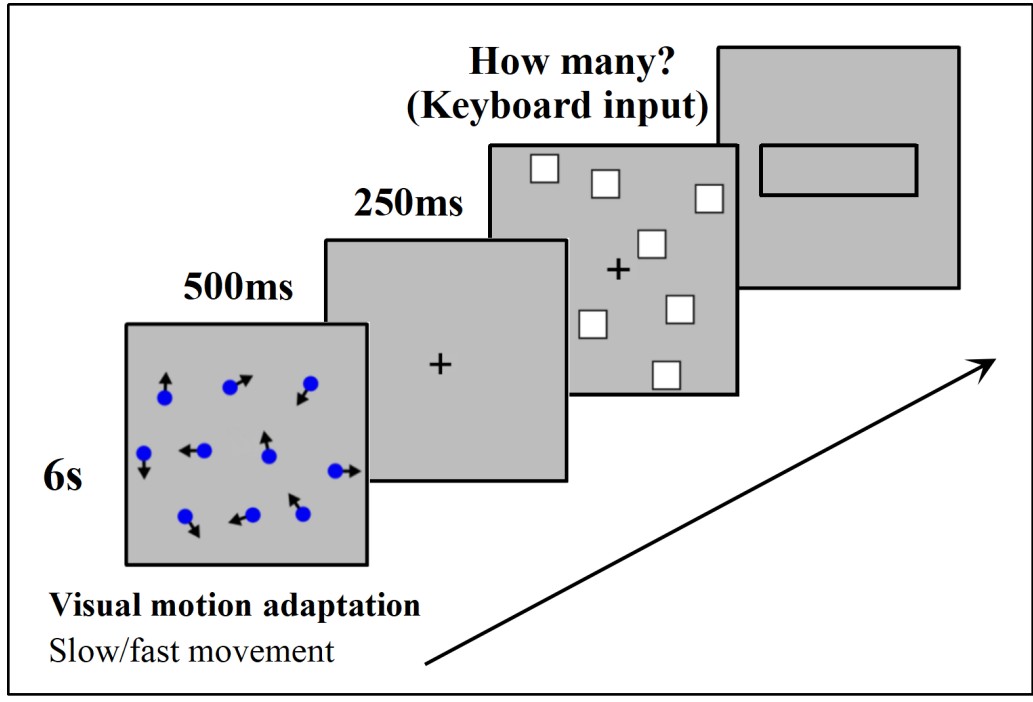

**Figure 4 Experimental procedure in detail.** A comprehensive overview of the steps taken during the study.

## Results

### Effects of adaptation

The results of Experiment 2 revealed that the adaptation index in the no-grouping condition was higher than in the grouping condition ($t = 2.072$, $p = 0.12$, Cohen's $d = 1.353$), indicating a stronger adaptation effect under no-grouping condition in visual motion adaptation (Fig. 5A). Regarding the perceived numerosity, Experiment 2 yielded similar results. Specifically, adaptation to fast visual motion (induced by viewing fast moving dots) led to underestimation of numerosity, led to a 14% reduction in apparent numerosity across all tested numerosities (linear regression slope of 0.85, compared to the baseline slope of 0.99), while adaptation to slow visual motion (induced by viewing slow moving dots) resulted in overestimation, adaptation resulted in a 5% increase in numerosity (linear regression slope of 1.04, compared to the baseline slope of 0.99), which is consistent to previous studies, these results were statistically significant and demonstrate a clear pattern (Fig. 5B).

### Groupitizing and adaptation effect

As a result of the ANOVA for CVs, we found that the main effect of grouping condition was borderline significant $F_{(1,23)} = 3.650$, $p = 0.069$. The sensory precision under the grouping condition was more accurate. A significant main effect of numerosity also emerged, $F_{(12,12)} = 35.734$, $p < 0.001$***. Similarly, the sensory precision of small numerosities was more accurate than that of large numerosities. And the interaction between group

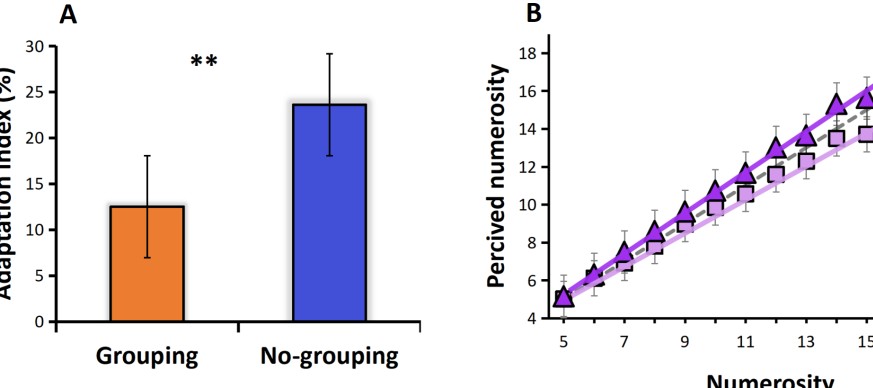

**Figure 5** (A) Adaptation index under grouping and no-grouping conditions. (B) The perceived numerosity, averaged across trials and subjects, varies as a function of physical numerosity in the three adaptation conditions. The analysis includes best-fitting linear regressions (R > 0.99 in all conditions). The regression slopes are as follows: no adaptation (dotted line) = 0.99; adaptation to fast visual motion (square) = 0.85; adaptation to slow visual motion (triangle) = 1.04 ($p < 0.001$ in both conditions).

conditions and numerosity was also significant, $F(12,12) = 3.995$, $p = 0.012^*$. The follow up simple interaction effect analysis revealed that, there is only a grouping effect in the intermediate numerosities (the grouping condition has more accurate sensing precision than the non-grouping condition), and there is no grouping effect in both the larger and smaller numerosities (Fig. 6).

For RTs, only the main effect of numerosity is significant, $F(12,12) = 8.294$, $p < 0.001^{***}$. Small numerosities also had shorter reaction times. And the interaction between numerosity and group conditions was borderline significant. $F(12,12) = 2.611$, $p = 0.55$.

### Discussion of Experiment 2

The findings from Experiment 2 closely mirrored those obtained in Experiment 1, revealing a consistent pattern of results. Specifically, in the condition of adaptation to visual motion, it was consistently observed that after adaptation to fast visual motion (*i.e.,* viewing the fast-moving lattice), the estimation of the number was consistently underestimated. Conversely, adaptation to slow visual motion (*i.e.,* viewing the slow-moving lattice) consistently resulted in overestimation of the number. These findings provide robust evidence for the stability of the adaption effect in both proprioceptive motor adaptation and adaptation to visual motion.

In Experiment 2′s visual motion adaptation process, we observed a lower adaptation effect in the grouping condition compared to the no-grouping condition (see Fig. 5A). This result suggests that the impact of visual motion adaptation on groupitizing is smaller than that of proprioceptive motor adaptation. Importantly, prior research emphasizes the positive impact of "groupitizing" on enhancing numerosity perception precision. In contrast, adaptation effects arise from perceptual distortions induced by adaptation, resulting in biases in numerosity estimation. Within the grouping condition, participants employed grouping strategies to actively mitigate perceptual biases induced by visual

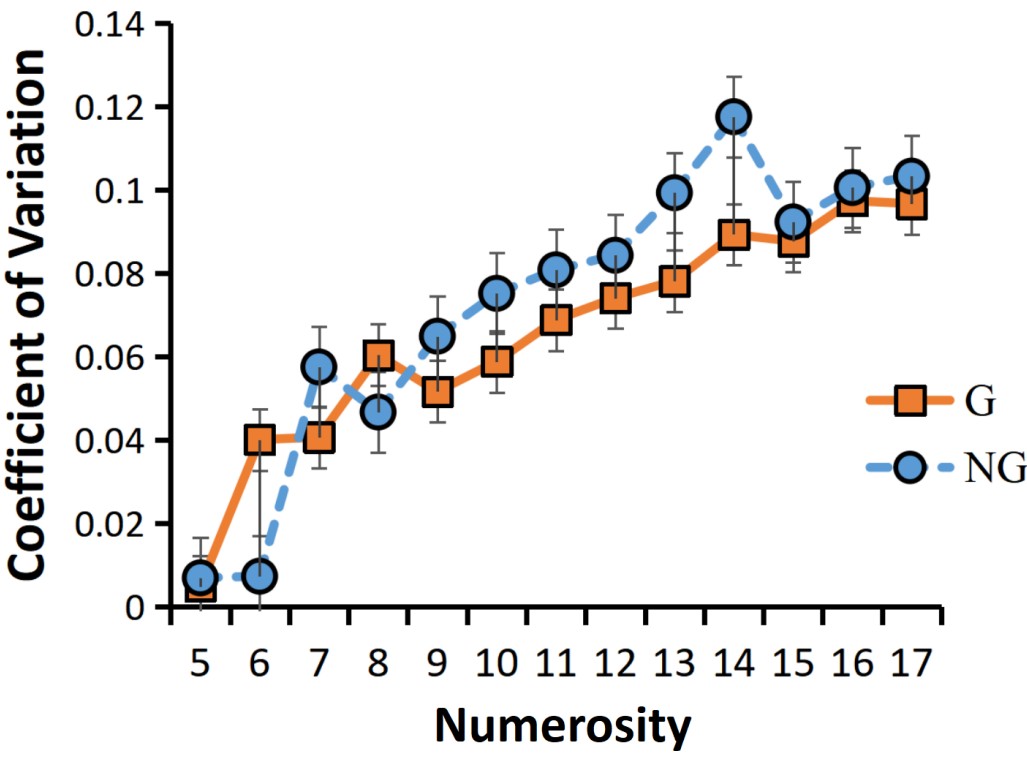

**Figure 6** **Interaction between numerosity and group conditions.** "G" for grouping condition, and "NG" for no-grouping condition.

motion adaptation. As a result, adaptation effects in the grouping condition consistently exhibited a lower impact compared to the no-grouping condition. This not only supports the positive impact of groupitizing on numerosity perception but also contributes to understanding how cognitive processes actively regulate the effects of perceptual adaptation on numerosity estimation. This emphasizes the dynamic interplay between cognitive strategies and visual adaptation, underscoring the role of grouping in fine-tuning the perceptual impact of adaptation.

Additionally, the impact of visual motion adaptation on groupitizing is observed to be smaller compared to proprioceptive motor adaptation, a finding that is further evident in the results of sensory precision. In Experiment 1, the grouping condition did not show a significant advantage over the no-grouping condition in terms of sensory precision, likely due to the effect of adaptation. However, in Experiment 2, the main effect of the grouping condition on sensory precision was borderline significant, indicating that the grouping condition after motion adaptation resulted in higher precision in the estimation task compared to the no-grouping condition. This provides further evidence that the use of a grouping strategy has a greater influence on visual motion adaptation compared to proprioceptive motor adaptation. It is noteworthy that the main effect of grouping condition on sensory precision was only borderline significant in this experiment, which may be attributed to the selection of numerosity range. As demonstrated in Fig. 6B,

sensory precision under the grouping condition with only intermediate values (9–14) was significantly higher than that under the non-grouping condition, with no significant difference observed between small and large values. These findings suggest that future experiments could explore the relationship between the adaptation effect and the grouping strategy by selecting intermediate values under the adaptation to visual motion condition to further investigate the adaption effect.

## GENERAL DISCUSSION

Based on the Gestalt theory of perceptual grouping, the phenomenon of "groupitizing" is considered a valuable strategy in numerosity perception, along with subitizing, counting, and estimation (*Anobile et al., 2020a*; *Anobile et al., 2020b*; *Ciccione & Dehaene, 2020*). Groupitizing involves perceiving visually grouped arrays as a single entity, which enhances the speed and precision of numerosity perception. The "A Theory of Magnitude" (ATOM) framework, suggests a close link between number and action. Motor adaptation, a cognitive process through which our motor system adjusts to changes in the environment or our own body, plays a role in numerosity perception by influencing how the brain processes numerical information and perceives numerosities (*Debats, Heuer & Kayser, 2023*; *Franklin et al., 2023*).

Previous studies have used motor adaptation technology to investigate the effects of motor adaptation on numerosity perception. It has been found that high-frequency motor adaptation leads to underestimation of numerosity perception, while low-frequency adaptation leads to overestimation. This phenomenon of motor adaptation, where prolonged exposure to a stimulus can result in perceptual changes, has been well-documented in numerosity perception tasks (*Anobile et al., 2016*; *Anobile et al., 2020a*; *Anobile et al., 2020b*). However, the interaction between motor adaptation and "groupitizing" in numerosity perception has not been thoroughly investigated. This raises the question of how groupitizing may interact with motor adaptation and affect numerosity perception.

The results of our study provide valuable insights into the interaction between motor adaptation and grouping strategies in numerosity estimation tasks. Experiment 1 replicated previous findings that adaptation to fast hand tapping led to an underestimation of numerosity, while adaptation to slow hand tapping resulted in overestimation, consistent with previous research (*Anobile et al., 2020a*; *Anobile et al., 2020b*; *Anobile et al., 2016*). However, contrary to previous findings, our results did not reveal a significant grouping effect on perception precision, indicating that motor adaptation may have modulated the grouping advantage. Interestingly, our data did show a grouping advantage in reaction time, with faster reaction times observed in the grouping condition compared to the no-grouping condition. This suggests that grouping may influence the speed of numerosity perception but not necessarily the precision.

However, surprisingly, adaptation effects persist even when presenting grouped stimuli after motor adaptation, showing a stronger impact under grouping conditions (Fig. 2A). This finding presents an intriguing and complex phenomenon. Firstly, previous

neuroscientific research suggests that motor adaptation induces neural adaptation to features such as motion direction and speed, thereby altering the responsiveness of the perceptual system (*Kohn & Movshon, 2004*). This adaptation may affect numerosity perception by reducing the sensitivity of the perceptual system to grouping information. Considering that neurons may have different functions in different tasks and conditions, motor adaptation and grouping effects may share some neural populations (*Kanai & Verstraten, 2005*; *Krekelberg, Boynton & Wezel, 2006*), leading to more intricate interactions between the two effects. Secondly, participants, when presented with grouped stimuli after motor adaptation, may choose to adjust their numerosity estimation strategy to adapt to changes in the perceptual system. This adjustment may involve balancing information from different sources to achieve optimal performance in numerosity perception tasks (*Wolfe, 2014*; *Summerfield & Egner, 2009*). To further unveil the neural basis of motor adaptation and grouping effects on numerosity perception, future research could employ neuroimaging techniques such as functional magnetic resonance imaging (fMRI) or electroencephalography (EEG). These techniques can provide more detailed insights into neural activity and connectivity patterns, allowing a deeper understanding of the neural mechanisms involved.

Experiment 2, focusing on visual motion adaptation, aimed to investigate whether adaptation to visual motion induces effects similar to proprioceptive motor adaptation and to examine the relationship between visual motion adaptation and groupitizing. The results closely mirrored those of Experiment 1, revealing a consistent pattern where fast visual motion adaptation led to underestimation, and slow visual motion adaptation led to overestimation. However, in Experiment 2, adaptation effects under the grouping condition were observed to be weaker compared to the no-grouping conditions. This difference suggests that the impact of visual motion adaptation on grouping effects might not be as pronounced as that of proprioceptive motion adaptation, providing room for participants to employ grouping strategies effectively and mitigate estimation biases. A plausible explanation for this observation is that participants may dynamically adjust their cognitive strategies under different adaptation conditions. In Experiment 1, the pronounced adaptation effects induced by proprioceptive motion adaptation may have inclined participants to preferentially adopt a singular adaptation strategy. Conversely, in Experiment 2, where visual motion adaptation was relatively weaker, participants seemed more inclined to flexibly adjust their cognitive strategies to adapt to grouping conditions.

Our results also suggest that the range of numerosity values used in the estimation task may influence the relationship between adaptation and grouping strategies. In Experiment 2, the borderline significant main effect of the grouping condition on sensory precision may be attributed to the limited range of numerosity values used (9–14). Future research could explore the relationship between adaptation and grouping strategies by manipulating the range of numerosity values under the adaptation to visual motion condition to further investigate the effects of grouping on sensory precision.

In conclusion, our research findings contribute to the understanding of the interaction between motion adaptation, groupitizing, and numerosity perception. The findings from Experiment 1 and Experiment 2 consistently showed that proprioceptive motor adaptation

involving high or low-frequency tapping of the fingers, as well as visual motion adaptation from observing rapid or slow-motion dot arrays, leads to subsequent underestimation or overestimation of presented numerosity. The use of a grouping strategy, known as "groupitizing", appeared to have a differential effect on motor adaptation depending on the context. In Experiment 1, the grouping strategy influenced proprioceptive motor adaptation, while in Experiment 2, it had limited effects on visual motion adaptation. Additionally, more research is needed to investigate the mechanisms of visual motion adaptation as it has received limited attention compared to proprioceptive motor adaptation in the field of motion perception. Overall, our study highlights the complex relationship between motor adaptation, grouping strategies, and numerosity perception and provides important insights for future research in this area.

### Funding

This research was funded by the National Natural Science Foundation of China (Research on the Cognitive Mechanism of the Groupitizing Strategies in Numerosity Perception; NO. 32360203). The funders had no role in study design, data collection and analysis, decision to publish, or preparation of the manuscript.

### Grant Disclosures

The following grant information was disclosed by the authors:
The National Natural Science Foundation of China (Research on the Cognitive Mechanism of the Groupitizing Strategies in Numerosity Perception): 32360203.

### Competing Interests

The authors declare there are no competing interests.

### Author Contributions

- Huanyu Yang conceived and designed the experiments, performed the experiments, analyzed the data, prepared figures and/or tables, authored or reviewed drafts of the article, and approved the final draft.
- Liangzhi Jia performed the experiments, prepared figures and/or tables, and approved the final draft.
- Jun Zhu performed the experiments, prepared figures and/or tables, and approved the final draft.
- Jian Zhang analyzed the data, prepared figures and/or tables, and approved the final draft.
- Mengmeng Li analyzed the data, prepared figures and/or tables, and approved the final draft.
- Chenli Li analyzed the data, prepared figures and/or tables, and approved the final draft.
- Yun Pan conceived and designed the experiments, authored or reviewed drafts of the article, and approved the final draft.
## Human Ethics

The following information was supplied relating to ethical approvals (i.e., approving body and any reference numbers):

All coauthors agreed with the contents of the manuscript. This study was performed in line with the principles of the Declaration of Helsinki. This study was approved by the School of Psychology Ethics Committee at Guizhou Normal University.

## Data Availability

The data is available at Zenodo: Yang, H., Jia, L., Jun, Z., Zhang, J., Li, M., Li, C., & Pan, Y. (2023). The interplay of motor adaptation and groupitizing in numerosity perception: Insights from visual motion adaptation and proprioceptive motor adaptation. Zenodo. https://doi.org/10.5281/zenodo.10421155.

The program that opens the original data is: E-Prime 3.0 https://pstnet.com/products/e-prime/.

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
