# Peer review of "The interplay of motor adaptation and groupitizing in numerosity perception: Insights from visual motion adaptation and proprioceptive motor adaptation"

_PeerJ, doi:10.7717/peerj.16887_

## Round 0.1 · original submission · Major Revisions

As you can see the Reviewers requested important clarifications and details about your experiments. I concur with this evaluation. Moreover, I believe the treatment of the literature may be improved in the Introduction and Discussion. For instance you mention that number sense is widespread in animals but you do not provide any reference for this. There are excellent recent reviews available, e.g.

- https://mitpress.mit.edu/author/andreas-nieder-28165/
- https://journals.sagepub.com/doi/full/10.1177/09637214221102146
- https://www.frontiersin.org/articles/10.3389/fpsyg.2021.641994/full
- https://www.frontiersin.org/articles/10.3389/fnana.2022.943504/full

Also, similarly when discussing ATOM some references to the biological literature could be useful, e.g.

- https://www.sciencedirect.com/science/article/pii/S0960982217313325
- https://www.cell.com/iscience/pdf/S2589-0042(20)30307-2.pdf
- https://www.cell.com/iscience/pdf/S2589-0042(23)02112-0.pdf

In any case, the final acceptance of this paper cannot be guaranteed at this stage. I would be pleased to receive a revised version that addresses all the issues raised by the Reviewers and I'll send the paper for further evaluation to the same Reviewers (if available).

Reviewer 1 ·

Basic reporting

In this study, the authors measured the effect of hand-motor adaptation and visual motion adaptation on spatial numerosity perception. The authors also measured how the strength of these adaptation would be altered by the spatial configuration of the stimuli to be estimated. The results of the motor adaptation replicate those already found in the literature and additionally show a modulatory effect of stimuli spatial configuration. Visual motion adaptation also demonstrated an effect on perceived numerosity but (if I understand correctly) this was independent from stimuli spatial configuration.

The study is interesting but suffers from major limitations that must be overcome before it can be fully evaluated. A critical aspect (to me) is the interpretation of the visual adaptation technique. This is defined as a 'passive viewing motor adaptation'. This sounds very strange to me. The participants were not presented with any stimulus related to the motor system but rather with a set of moving dots. Even in the abstract it is named as 'visual feedback motor adaptation', but it is not motor but purely visual. This issue in my opinion needs to be seriously reconsidered. Furthermore, the authors proposed that the results obtained with motor and visual adaptation are similar but to me they look very different. In the visual motion adaptation exp, the numerosity of the adapting stimulus was not varied and the results obtained are contrary to those predicted and usually obtained with numerosity adaptation techniques (detailed below).

The methods lack clarity. The discussions are vague, unfocused, and considering visual adaptation as a motor adaptation. The figures are unclear in what they show. The data analysis contains errors. The references are often wrong in format and the English needs revision. In brief, before a substantial revision is impossible to me to fully evaluate the work and its impact. In any case, I list below the weaknesses that are most evident to me, hoping that these will be helpful to improve the ms.

Experimental design

EXPERIMENT 1:

Methods:
• After motor adaptation, where was the visual stimulus was presented? Near the adapted area? This is a rather crucial point. The effect under investigation has in fact been shown to be spatially selective (Anobile et al. 2020, Distortions of visual time induced by motor adaptation, J Exp Psychol ). Please specify at what distance from the adapted area the visual stimulus was presented.

• Were the grouped and random conditions tested in separate blocks?

Data analysis:
• How was the responses bias induced by adaptation measured? The authors report that: "In line with previous studies (Anobile et al., 2020), the total adaptation effect was calculated as the normalised difference between the coefficients of variation (CVs) obtained from the two adaptation levels: high and low frequency tapping, viewing fast- or slow-moving dots." This method does not calculate the effect of adaptation (which is related to accuracy), it is an index of groupitizing advantage on sensory precision. In summary, as presented, the analyses do not allow me to understand how the main parameter (adaptation effect) has been calculated.


Results: Effects of adaptation
• The results begin by referencing Figure 3A (line 216) but describing other data, probably related to Figure 2?. Has been reported that “the adaptation effect observed under the grouping condition was found to be significantly stronger than that under the no-grouping condition (t = 3.808, p = 0.002, Cohen's d = 1.056)”. However Figure 2 shows the opposite. What is wrong here? The graph or the text? What does panel B of fig 2 show? The data obtained in the random or grouped condition?.

Discussion:
• Line 262 "perception accuracy", this is not accuracy, it is precision. In general, these two parameters are confused along all the ms.


EXPERIMENT 2:

Methods:
• If I understand correctly, the numerosity of the adaptor was set at N10 while the test stimuli ranged from 5 to 17. This, according to adaptation should have led to an overestimation of test numerosities higher than N10 and an underestimation of numerosity below N10. Contrarily to this prediction, the results show an overall overestimation in the slow-motion condition and an underestimation in the fast-motion condition. How do these results reconcile with those predicted by numerosity adaptation? These results are counter-intuitive and need to be investigated further, e.g. by adding a condition with an adapter more numerous than the highest tested numerosity.

• Please specify the visual motion speed, now is reported as 'fast' and 'slow'. Why adapting and testing dots were a different colour? It has been shown that when adaptor and stimulus do not share the same item’s colour, numerosity adaptation effect significantly decreases compared to when they share the same colour (Grasso et al, 2022. Numerosity perception is tuned to salient environmental features. iScience). Could this have influenced the current results?

Results:
• Figure 4: The figure states "motor adaptation" but it is visual adaptation to movement, not motor. Panel B of fig 4 reports what? The data obtained in the random or grouped condition or the average of the two? If the idea is to compare the effect between grouped and ungrouped stimuli, these should be shown individually.

Discussion:
• Related to the issue raised above, the authors suggest that the findings from Experiment 2 closely mirrored those obtained in Experiment 1, but to me this is not the case. Experiment 1 manipulated the numerosity of the adaptation actions, obtaining the expected result (overestimation after a few actions, underestimation after many actions) whereas in Experiment 2, always adapting to a numerosity intermediate to the range tested, constant overestimation and underestimation was found and not predicted by what is expected from adaptation (see above).

• Here and in other sections the authors refer to motion adaptation as “low-frequency and high-frequency adaptation” but there is no frequency modulation, what has been modulated is the visual motion speed.

General discussion
It is unclear to me how the results can be explained by the Predictive Coding framework. What links motor adaptation to visual grouping from a predictive coding perspective? If the prior is a sequence of actions, what information should it provide regarding the spatial configuration of the next visual stimulus? In general, as the results are still unclear to me, the discussions need further evaluation after ms revision.

Minor comments:
Please check references format along all the ms. Here some example of errors:
Line 91; (P. A. Maldonado Moscoso et al., 2020…) "P.A." should be eliminated.
Lines 94 e 95: (Anobile & Castaldi et al., 2021), (Anobile & Arrighi et al.,)… "&" should be eliminated.
Line 383. The Statement section is reported before the discussions. Is this an error?

Validity of the findings

Given the lack of clarity on several aspects (detailed above), it is currently difficult for me to assess the validity of the results.

Reviewer 2 ·

Basic reporting

In this study Huanyu et al. investigate the effects of motor (self-produced movements; Exp 1) and sensory adaptation (moving dots display; Exp 2) on numerosity perception of grouped and ungrouped stimuli. The results of Exp 1 allegedly (see below) replicated previous findings showing that, as a consequence of adaptation to numerous self-produced motor routines, perceived numerosity of the following visual stimuli is underestimated while the opposite occurred as a consequence for adaptation to few motor routines. The results of Exp 2 extend these findings to indicate that similar distortions in perceived numerosity also occur for adaptation to fast or slow visual motion.

This study is interesting as it tackles on intriguing issue as the interaction of several contextual effects of numerosity perception (i.g. motor or visual adaptation) with the phenomenon of groupitizing, interaction that has never been investigated before. However, the study also suffers of several major limitations such as a poor description of the experimental paradigms, a poor description and visualization of the achieved results, a lack of clarity of the exploited terminology; all issues that make the current version of the manuscript not eligible of publication on PeerJ. Please, find below a list of all the concerns the Authors should deal with in case they want to prepare a revised version of the manuscript.

Experimental design

Exp 1: The goal of experiment 1 is to leverage on the technique of motor adaptation to investigate whether it affects numerosity perception differently for grouped and ungrouped stimuli. However, from the description of the experimental methods it is not clear where the tapping routines would occur. This is important as previous experiments have revealed that the spatial congruency between the area where the motor routines take place and the position of the subsequent visual stimuli is of critical importance to achieve motor adaptation aftereffects (see for example Anobile et al 2020). What was the spatial correspondence between the visual stimuli and the tapping area in the present study?
The description of the procedures makes clear that “The two adaptation levels were tested separately” but what about the two spatial arrangements of the visual stimuli? Were trials with grouped and ungrouped stimuli tested in separate sessions as well?

Validity of the findings

Results Exp1:
This is the session I have found to be the most confusing. The right-hand panel of Figure 2 depicts data about how motor adaptation distorts perceived numerosity for all different tested numerosities. On the contrary to what is described in the methods and reported in the panel on the left, these results are about the accuracy of the estimates (difference between the reported and the veridical numerosity) and they have nothing to do with the idea of precision that allegedly the authors measured via the Coefficient of Variation index. In light of this, what does the term “adaptation effect” indicated on the ordinate of the left-hand panel of Fig 2 stand for? If this adaptation effect is related to the difference in coefficient of variation (precision), then the right panel of Fig 2 needs to be changed with something similar to the right panel of Figure 6. Hoever, if the data really refer to measures of precision, please nte that the rpeswente results cannot be compared with those reported by Anobile et al. In particular it does not make sense the sentence in which the Authors declare that “In line with previous studies (Anobile et al., 2020), the total adaptation effect was calculated as the normalized difference between the coefficients of variation (CVs) obtained from the two adaptation levels”. What is the study the Authors refer to here? If they refer to the study by Anobile and coll. published on JEP (Distortions of visual time induced by motor adaptation), the sentence is wrong because in that study, the Authors were measuring adaptation effect in terms of differences of Point of Subjective Equality (PSEs), that is they were measuring changes in accuracy, not precision (see equation 1). If instead they refer to the study “Groupitizing: a strategy for numerosity estimation” the sentence is anyway wrong because in this latter study, no adaptation technique was exploited.

Discussion of the results of Exp1:
The confusion between accuracy and precisin obviously affect also the section dedicated to the discussion of the results of Exp 1. The Authors discuss changes in perceived numerosity induced by motor adaptation such as over or underestimation (that is changes in accuracy of the estimates) with data about adaptation effects measured in terms of precision (that is based on CVs, see equation 2) to make the whole discussion almost not comprehensible.

Additional comments

There are also several issues about Exp 2.
Exp 2 seems to be motivated by the idea to investigate the effects of adaptation to visual motion on numerosity perception as, quoting by the manuscript “[it is] unclear whether passive viewing of fast or slow moving objects elicits similar estimation biases in subsequently presented numerosities.” Indeed a previous study directly tackled the effect of adaptation to fast or slow visual patterns on numerosity perception and also took into account different motion profiles (translational, rotational and radial motion; see Motion-induced compression of perceived numerosity, Fornaciai et al. Sci Rep 2018). Given that, the Authors should better motivate the rationale of study 2 to stress out that they add on the previous literature the effect of motion adaptation on GROUPED stimuli and - of course - discuss the present results in light of those reported by Fornaciai and colleagues.

Methods Exp 2:
Another issue is about the adapting stimuli exploited in Exp 2. The adaptor is a display of moving dots so why should it be defined as “passive viewing motor adaptation paradigm”? What motor stand for here? I would clearly define this adaptation condition as “adaptation to visual motion”

Results Exp 2:
Again, the results of Exp 2 are difficult to evaluate as we have a figure (5) reporting changes in accuracy induced by adaptation to visual motion and then in Fig 6 we are presented with data about precision (CVs). All of this must be clarified to provide a clear take home message to the readers.

A final comment. The data on the left-hand panel of Figure 6 are absolutely not clear. Apart from the wrong labes (adaptation TAPE and Numeroisty), it looks like that in the “proprioceptive” (motor?) condition (bars on the left in panel A) there is no difference in precision between the grouped and the ungrouped condition. However, if I got it right, these data should be the same as those reported in panel A of Figure 2 where there was a robust difference between the grouped and ungrouped. Did the authors mistake the two labels -motor/proprioceptive and visual feedback- conditions here? In any case the terms visual feedback is wrong as no feedback was provided to the participants. It is again just a condition in which participants were adapted to visual motion!

Minor points:
Most of the references need to be amended as the formatting is wrong. For example, the references “Anobile & Arrighi et al. 2021”, “Anobile & Castaldi et al., 2021”, Anobile & Domenici et al., 2020 need to be edited to remove the character &
The reference “P. A. Maldonado Moscoso et al., 2020” should be read Maldonado Moscoso et al., 2020

More, in the reference list there are duplicates:
Anobile, G., Arrighi, R., Togoli, I., & Burr, D. C. (2016). A shared numerical representation for action and perception. ELife, 5 http://doi.org/10.7554/eLife.16161
Anobile, G., Arrighi, R., Togoli, I., & Burr, D. C. (2016). A shared numerical representation for action and perception [Journal Article; Research Support, Non-U.S. Gov't]. ELife, 5 http://doi.org/10.7554/eLife.16161

Togoli, I., Crollen, V., Arrighi, R., & Collignon, O. (2020a). The shared numerical representation for action and perception develops independently from vision. Cortex, 129, 436-445. http://doi.org/10.1016/j.cortex.2020.05.004
Togoli, I., Crollen, V., Arrighi, R., & Collignon, O. (2020b). The shared numerical representation for action and perception develops independently from vision. Cortex, 129, 436-445. http://doi.org/https://doi.org/10.1016/j.cortex.2020.05.004

Reviewer 4 ·

Basic reporting

Clarity of Research
Objectives: While the abstract and introduction mention that the study aims to explore the effects of visual feedback motor adaptation, the methods and results seem to focus primarily on proprioceptive motor adaptation. This discrepancy needs to be clarified. Are both types of motor adaptation considered in the study?

Methodological Details: The paper is light on some details crucial for reproducibility. The technology used for recording hand movement and visual stimuli could be more explicitly stated. Additionally, the inclusion criteria for participants remain vague.

Statistical Methodology: It might be helpful to provide more information on why a 2 x 2 x 13 repeated measures ANOVA was specifically chosen for this analysis. How do the authors justify treating the three factors as equally important for the analysis?

Experimental design

No comments.

Validity of the findings

No comments.

Additional comments

No comments.

---

## Round 0.2 · accepted · Accept

The Reviewers are satisfied with your revision, and I am thus happy to accept the paper.

Reviewer 1 ·

Basic reporting

The ms is improved and I think I now ready for publication.

I have just some minor suggestions:

I am not sure that “proprioceptive motor adaptation” is the best definition. I understand that tapping implies proprioceptive signals but paraphs “motor number adaptation” should make the message clearer and (probably) more directly linked to the existing literature. Obviously, this choice is up to the authors. A have a similar advice for “motor adaptation technology”. I think it could be better mentioned as an” experimental paradigm” rather than a “technology”.

Line 105: “robust association between motion…”. As suggested in the previous round of revision, “motion” is confusing, please refer to action (e.g. “robust association between action and numerosity perception”).

Figure 3. Axis label: Adaption should be Adaptation.

Experimental design

no further comments

Validity of the findings

no further comments

Additional comments

no further comments

Reviewer 2 ·

Basic reporting

no comment

Experimental design

no comment

Validity of the findings

no comment

Additional comments

The Authors did a great job in addressing all the issues I had highlighted about the previous version of the manuscript. Indeed find the revised version of the paper much improved especially in terms of clarity about the dimensions (accuracy vs precision) that have been actually investigated. However, I feel like suggesting the Authors to go throughout the manuscript a very last time for a final proofread as some typos might be still be around. For example, I have noticed that the world "adaption" is used in place of "adaptation" several time.

Reviewer 4 ·

Basic reporting

The authors have addressed all my concerns.

Experimental design

No comments.

Validity of the findings

No comments.

Additional comments

No comments.